# Nanotubes: Carbon-Based Fibers and Bacterial Nano-Conduits Both Arousing a Global Interest and Conflicting Opinions

**Silvana Alfei** [1,*] and **Gian Carlo Schito** [2]

1   Department of Pharmacy (DIFAR), University of Genoa, Viale Cembrano, 4, 16148 Genoa, Italy
2   Department of Surgical Sciences and Integrated Diagnostics (DISC), University of Genoa, Viale Benedetto XV, 6, 16132 Genova, Italy
*   Correspondence: alfei@difar.unige.it; Tel.: +39-010-355-2296

**Abstract:** Nanotubes (NTs) are mainly known as materials made from various substances, such as carbon, boron, or silicon, which share a nanosized tube-like structure. Among them, carbon-based NTs (CNTs) are the most researched group. CNTs, due to their nonpareil electrical, mechanical, and optical properties, can provide tremendous achievements in several fields of nanotechnology. Unfortunately, the high costs of production and the lack of unequivocally reliable toxicity data still prohibit their extensive application. In the last decade, a significant number of intriguing nanotubes-like structures were identified in bacteria (BNTs). The majority of experts define BNTs as membranous intercellular bridges that connect neighboring bacterial cell lying in proximity. Despite recent contrasting findings, most evidence suggested that bacteria exploit NTs to realize both antagonistic and cooperative intercellular exchanges of cytoplasmic molecules and nutrients. Among other consequences, it has been proposed that such molecular trade, including even plasmids, can facilitate the emergence of new non-heritable phenotypes and characteristics in multicellular bacterial communities, including resistance to antibiotics, with effects of paramount importance on global health. Here, we provide an enthralling comparison between CNTs, which are synthetically producible and ubiquitously exploitable for improving the quality of human life, and BNTs biosynthetically produced by prokaryotes, whose functions are not still fully clarified, but whose greater knowledge could be crucial to better understand the mechanisms of pathogenesis and combat the phenomenon of resistance.

**Keywords:** carbon nanotubes (CNTs); bacterial nanotubes (BNTs); mechanical prowess; electrical potency; optical properties; CNTs biomedical applications; cytoplasmic molecular trade; plasmids trade; non-heritable phenotypes; BNTs as expression of bacterial death



## 1. Introduction

Nanotechnology is an area of research and manufacturing technology of global interest, which deals with a variety of materials produced at a nanometer scale (<100 nm) through different chemical and physical methods [1]. Nanotubes (NTs) belong to a promising group of nanomaterials which allows approaching several new electronic, magnetic, optical, and mechanical properties [2].

Even though many NTs containing boron, silicon, and molybdenum have been extensively studied, currently, carbon nanotubes (CNTs) are the most researched group among existing NTs structure [2]. Indeed, as it was evidenced by a search made on Scopus by using as keywords first "nanotubes" and then "carbon nanotubes", in the last two decades (2001–2021) up to 182,033 documents concern nanotubes of which 145,162 (80%) concern carbon nanotubes (Figure 1).

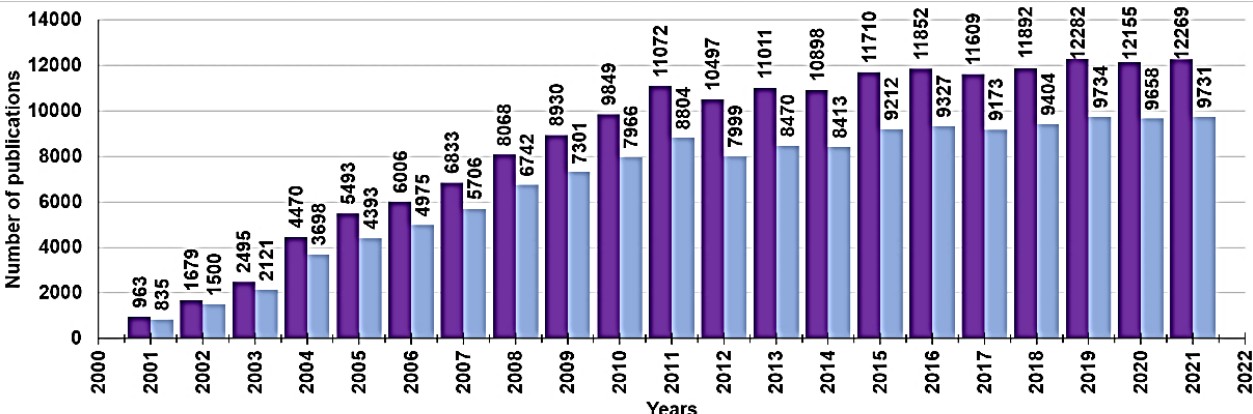

**Figure 1.** Number of publications per year during the last 20 years according to Scopus, concerning nanotubes (purple bars) and carbon nanotubes (light purple bars).

CNTs are composed of graphite sheets rolled up in an unbreakable and non-stop hexagonal-like lattice structure, in which carbon atoms appear at the tops of the hexagon-type forms. Based on the number of carbon sheets, CNTs are categorized as single wall carbon nanotubes (SWCNTs), double wall carbon nanotubes (DWCNTs), and multi-wall carbon nanotubes (MWCNTs) [2]. CNTs belonging to all categories vary by purity, length, and functionality, but possess a plethora of properties, including high electrical conductivity, high tensile strength, light weight [2–5], high biocompatibility [6], capability of molecules immobilization for their transportation, large surface area, chemical inertness, possible presence of functional groups, elasticity, thermal conductivity and expansion, electron emission capacity, and high aspect ratio [2–5].

The unique composition, geometry, and properties of CNTs enable numerous potential applications. A great challenge of researchers in the field consists of reducing costs of CNTs production down to commercially viable levels, and it seems that scaling up in their synthesis is happening. CNTs can be applied for energy storage, biomedical uses, air, and water filtration. Additionally, CNTs can be used as molecular electronics, thermal materials, structural materials, electrical conductors, fabrics and fibers, catalyst supports, conductive plastics, conductive adhesives, as well as ceramics [2].

Interestingly, CNTs have great potential to be applied in nanomedicine for disease diagnosis and drug targeting, as well as to transport various biomolecules, such as proteins, DNA, RNA, immune-active compounds, and lectins [7,8].

CNTs are also extensively applied to develop electrochemical sensors, DNA-based sensors, as well as piezoelectric and gas sensors [2]. Additionally, opportunely structured CNTs, such as cationic CNTs have revealed to possess interesting antibacterial and anti-fungal activity [2]. As an example, Table 1 summarizes the several applications that CNTs could have only in the biomedical area.

**Table 1.** Main biomedical application of CNTs.

| Applications | | Principle | Detection Target | Year Ref. |
|---|---|---|---|---|
| Sensors | Electrochemical sensors | Made of silver nanoparticles (AuNPs) + MWCNTs Mannan-Os Adducts | Dopamine | 2015 [9] |
| | | Made of glassy carbon electrodes modified with MWCNTs and copper microparticles (CuMPs) dispersed in polyethylenimine (PEIs) | Amino acids and glucose | 2016 [10] |
| | | Based on MWCNTs | Clostridium tetani | 2016 [11] |
| | | Based on AgNPs/bismuth (Bi)NPs/MWCNTs nafion modified | Lead and Cadmium | 2020 [12] |
| | | Based on carbon sensor fabricated with coalesced ruthenium-doped titanium dioxide (Ru–TiO$_2$) NPs and MWCNTs | Cetrizine | 2019 [13] |
| | | Based on glassy carbon sensor modified with MWCNT in pH 9.0 phosphate-buffered solution (PBS) | Methdilazine | 2019 [14] |
| | | Based on (Ru–TiO$_2$) NPs and MWCNTs | Flufenamic acid (FFA) Mefenamic acid (MFA) | 2019 [15] |
| | | Based on BiNPs decorated MWCNTs modified carbon paste electrode (Bi-MWCNT/MCPE) at physiological pH | Gallic Acid (GA) | 2020 [16] |
| | | | **Function** | |
| | Piezoelectric sensors | Based on MWCNTs on polydimethylsiloxane (PDMS) | N.R. | 2017 [17] |
| | | Based on graphene, CNTs and graphene-CNTs composite | N.R. | 2018 [18] |
| | | Based on MWCNT on PDMS as substrate. | For developing robotic hands (rehabilitation) Strain detecting needle for tissue characterization | 2019 [19] |
| | | Based on MWCNTs on thermoplastic urethane as substrate. | Sensors integrated in gloves and bandages for assessing specific human functions | 2019 [20] |
| | | Based on MWCNTs on PDMS substrate | To measure pressure directly without the use of deformation materials. | 2019 [21] |
| | | | **Detection Target** | |
| | Gas sensors | Based on a resonant electromagnetic transducer in microstrip technology | Volatile Organic Compounds (VOCs) | 2019 [22] |
| | | Based on dye functionalized matrix anchored onto MWCNTs. ammonia | Sulphur dioxide and chlorine | 2018 [23] |
| | | Based on tungsten oxide nano-bricks and MWCNTs | Ammonia gas | 2019 [24] |

**Table 1.** *Cont.*

| Applications | Principle | Detection Target | Year Ref. |
|---|---|---|---|
| | **Transported Drugs** | | |
| Drug Targeting | Based on MWCNT and pH responsive gel consisting of chitosan coated magnetic nanocomposites | Doxorubicin (DOX) to U-87 Glioblastoma cells | 2019 [25] |
| | Based on stimuli responsive CNTs using Ag nanowires to stimulate the drug release from the core of NTs | Cisplatin | 2017 [26] |
| | Based on electro responsive polymer-MWCNT hybrid hydrogel | Sucrose | 2013 [27] |
| | Based on MWCNTs biodegradable, biocompatible nanocomposite hydrogel. | Diclofenac sodium | 2016 [28] |
| Cancer diagnosis and treatment | Based on methotrexate loaded MWCNTs releasing drugs by enzymatic cleavage | Methotrexate to in vitro breast cells | 2010 [29] |
| | Based on DOX loaded dendrimer modified MWCNTs releasing drugs at low pH | DOX | 2013 [30] |
| | Based on cationic MWCNTs-NH3þ for direct intertumoral injections | Apoptotic siRNA against polo-like kinase (siPLK1) in calu6 tumor xenografts | 2015 [31] |
| | **Target bacteria/Applications** | | |
| Antibacterial agents | Based on chitosan/MWCNTs nanocomposites | *Enterococcus faecalis* *Staphylococcus epidermidis* *Escherichia coli* | [32–38] |
| | Based on iron oxides/SWCNTs | *E. coli* | 2018 [39] |
| | Based on Ag–$Fe_3O_4$/SWCNTs | N.R. | 2018 [40] |
| | Based on cyclodextrin/Ag/MWCNTs | N.R. | 2014 [41] |
| | Based on MWCNTs containing carboxylic functions | N.R. | 2015 [42] |
| | Based on polyaniline/graphene/CNTs | *Staphylococcus aureus, E. coli* | 2018,2012 [43,44] |

**Table 1.** *Cont.*

| Applications | Principle | Detection Target | Year Ref. |
|---|---|---|---|
| | Based on MWCNTs | Treatment of drinking water through removal and inactivation of virus and bacteria | 2011 [45] |
| | Based on MWCNTs functionalized with mono-, di-, and tri-ethanolamine | *E. coli, Klebsiella pneumonia* *Pseudomonas aeruginosa* *Salmonella typhimurium, Bacillus subtilis* *S. aureus, Bacillus cereus* *Streptococcus pneumonia* | 2014 [46] |
| | Based on dispersion SWCNTs/tetra-arylbimesityl derivative activated with carboxy groups | *E. coli, S. aureus* | 2019 [47] |
| | **Target fungi** | | |
| | Based on chitosan/MWCNTs | *Aspergillus niger, Candida trobicalis* *C. neoformans* | 2000 [48] |
| Antifungal agents | Based on functionalized CNTs | *A. niger, C. albicans, A. fumigatus,* *Penicillium chrysogenum* *Saccharomyces cerevisiae, Fusarium culmorum* *Microsporum canis, Trichophyton mentagrophytes* *T. rubrum, P. lilacinum* | 2013 [49] |
| | Based on dispersion SWCNTs/tetra-arylbimesityl derivative activated with carboxy groups | *C. albicans* | 2019 [47] |

A multitude of other possible applications for CNTs exist, as solar collectors, catalyst supports, nano-porous filters, and coatings of all types. Surely, many unexpected applications for these amazing materials will occur in the next years, which will confirm how CNTs could be the most important and valuable nanomaterial ever used. Several experts in the field are thinking to use CNTs for obtaining conductive and/or waterproof paper. CNTs possess the capability to absorb infrared light and may be applied in the I/R Optics Industry. To avoid tedious repetitions, more information and more exhaustive literature data concerning the potential applications of CNTs will be provided in the following parts of this article. Intriguingly, nanostructures referred to as nanotubes exist also in the world of living beings. In eukaryotic cells, nanotubes promote intercellular transport of cytoplasmic substances, organelles, membrane constituents, as well as pathogenic microorganisms, including viruses [50]. Additionally, even if contrasting opinions and findings are emerged recently, it was reported that bacterial cell–cell interactions are mediated by tubular protuberances, labeled nanotubes, by similarity to their eukaryotic counterparts [51]. Indeed, it has been reported that bacteria, in addition to keep complex molecular exchanges and correspondence with neighboring eukaryotic and prokaryotic cells, by specific apparatuses including type III, IV, and VI secretion systems [52,53], they also mediate contact-dependent interactions by nano-membranous tubular conduits (namely nanotubes), which bridge neighbouring or distant cells of same or different species [51,54].

First observed in *B. subtilis* and *E. coli*, according to the most studies existing in the literature, bacterial nanotubes (BNTs) are hollow tube-shaped nanostructures made of membranous material connecting cell pairs that allow for the sharing of cytoplasmic metabolites, nutrients, toxins, cytoplasmic proteins, and even plasmids. Although the implication of BNTs in the exchange of cytoplasmic material has been recognized in several bacteria of non-marine species, and mainly when they are grown on solid surfaces or in biofilm assemblies [55–75], their occurrence and potential ecological significance in marine bacteria has been only recently reported by Patel et al. [58]. Particularly, these researchers identified tube-like structure, such as those detected in *B. subtilis* in marine isolates of *Pseudoalteromonas* sp. TW7 and *Alteromonas* sp. ALTSIO. According to what reported by Dubey et al., which in their study elucidated bacterial nanotubes (BNTs) architecture, dynamics, and molecular components, utilizing *B. subtilis* as a model organism, BNTs exhibit remarkable complexity and are formed exclusively by intact living cells [53]. The authors evidenced that BNTs exist as both intercellular tubes and as elongated extending tubes, surrounding the cells in a "root-like" fashion, which markedly expand the cell surface and should serve to bacteria mainly to obtain nutrients [54]. These structures are formed by bacteria in minutes, displaying rapid movements. BNTs are composed of chains of membranous lipid portions encompassing an uninterrupted lumen. Furthermore, it was found that in BNTs is present a conserved calcineurin-like phosphodiesterase protein, namely YmdB, retained essential for both nanotube production and intercellular molecular trade [54]. More recently, other components enabling the biogenesis of BNTs, have been identified in a molecular apparatus (the flagellar body proteins) which provides a platform for nanotube biogenesis [59]. Moreover, although several genes involved in the BNTs formation had been reported, the pivotal role of a specific sigma factor and of some autolysins in BNTs formation was very recently investigated and elucidated in *B. subtilis* [60]. Interestingly, other functions of BNTs, different from those previously established, the rarity in their formations and their production by dying or even dead cells, have been described in this recent study [60]. It is curious that structures so different and with so different functions and/or applications share the same name, arouse conflicting opinions, and need both of further extensive study to clarify many outstanding questions, here we provided an intriguing comparison between CNTs, which are the most researched and applied group of nanotubes synthetically producible by man from inert material and whose actual environmental impact is highly debated, and the nanotubes biosynthetically produced by bacteria, on which, as evidenced in the present work, a plethora of conflicting findings exists.

## 2. Approaching CNTs: Properties, Uses and Current Hurdles

Carbon is a chemical element owning atomic number 6 and is one of the most abundant elements in the Universe by mass. Carbon can form almost 10 million of different pure organic compounds having an uncommon ability to form polymers at the temperatures commonly present on earth, thus resulting the chemical basis of all known life [61].

CNTs are thin and long cylinders made of carbon, which were found for the first time in 1991 by Sumio Iijima [62]. Although many physical properties of CNTs are still obscure and need to be disputed, it is known that CNTs possess a plethora of thermal, electronic, mechanical, and structural properties that can vary based on the different kinds of NT. In this regard, CNTs differ mainly for their diameter, length, chirality, or twist [61]. Table 2 collects the key properties and potential uses of CNTs, as well as the current remaining technical and not technical hurdles, which still limit the CNTs' success and their extensive application. Note that the biomedical applications of CNTs have been only superficially mentioned in Table 2 because already extensively reviewed in the previous section (Table 1). Note that the biomedical applications of CNTs have been only superficially mentioned in Table 2 because they were already extensively reviewed in the previous section (Table 1).

**Table 2.** Key properties and potential uses of CNTs, as well as the residual current hurdles which limit CNTs extensive applications.

| Key Properties | Potential Uses | Current Hurdles |
|---|---|---|
| **Size [61]** | **Nano-electronics [64]** | Electronic heterogeneity [76] |
| | | Crystallographic defects |
| | | Stone–Wales defects |
| Ø < 100 nm<br>Thickness = 1–2 nm<br>Ideally infinite length | Conductors (SWCNTs)<br>Superconductors<br>Semi-conductors [1]<br>Supercapacitors | Not tunable conductivity |
| **Electrical and electrochemical properties** | **Interconnects** | ⇑ Controllable orientation |
| ⇑ Electrical conductivity<br>Constant resistively [63]<br>Electrons emission capacity | Chips manufacture [65]<br>Conductivity-enhancing components | ⇓ Organized configuration |
| **Thermal Properties** | | ⇑ Variation in size and density [76] |
| ⇑ Heat conductivity<br>Expansion | **Optics and photonics** | |
| **Mechanical properties** | Light-emitting diodes (LEDs) [66,67] [2] | |
| ⇑ Tensile strength<br>⇑ Elastic modulus | Photodetectors [68] [2] | |
| **Optical Properties** | Bolometers [69] [2] | Export restrictions |
| Absorption properties | | |
| Photoluminescence (fluorescence) | Optoelectronic memory devices [70] [2] | |
| Raman spectroscopy properties | | |

**Table 2.** *Cont.*

| Key Properties | Potential Uses | Current Hurdles |
|---|---|---|
| **Others** | **Batteries [71]** | |
| Easily modifiable structure | | |
| Presence of functional groups | Supercapacitors production | |
| Chemical inertness | Ultra-thin flexible batteries<br>Implantable medical devices | |
| | **Cleaning up polluted environments** | |
| | Water filtration<br>Air filters, such as smokestacks [72] | |
| Easily optimizable solubility and dispersion | **Others** | Environmental concerns [76] |
| | Transistors production [73,74] | |
| | Energy production | |
| | Solar cells [75] | |
| Light weight | Energy Storage | |
| ⇧ Biocompatibility | CNT-based fibers and fabrics | |
| Capability of molecules immobilization | CNT-based ceramics | |
| Transport of protein, DNA, RNA | | |
| Large surface area | Sources of light | |
| High capability of absorbing chemicals from their surroundings | | Entrenched dominance of other material |

[1] Conductive capacity of CNTs depends on the chirality degree, i.e., the twist degree and dimension of the diameter of the actual nanotube; [2] SWCNTs; ⇧ = high, wide; ⇩ = scarcely.

## 2.1. Impediments to the Extensive Application of CNTs

### 2.1.1. Environmental and Human Safety: Some Case Studies

The National Institute for Occupational Safety and Health (NIOSH) is the most important federal agency in the United States which conducts research and provides supervision on the occupational safety and on the consequences for human health which could be derived from the applications of nanomaterials. Studies have revealed that NPs may pose a greater health risk than bulk materials due to the greater surface area/unit mass ratio. In this regard, although continuous toxicological studies are ongoing, the biological interactions of NPs, including CNTs, are not well understood. Many are the factors which can create confusion. Additionally, since carbon is practically biologically inert, part of the toxicity attributed to CNTs could depend on the residual metal catalyst used for their preparation rather than on CNTs themselves. Promisingly, only Mitsui-7 has been unequivocally confirmed to be carcinogenic, so far [77]. Fortunately, differently from many common mineral fibers, such as asbestos, most SWCNTs and MWCNTs do not possess the requisites, such as size and aspect-ratio, to be classified as respirable fibers, thus limiting their possible toxicity. In 2013, NIOSH published a Current Intelligence Bulletin detailing the potential hazards and recommending the exposure limit for CNTs and fibers [78]. In October 2016, SWCNTs have been registered across the European Union's Registration, Evaluation, Authorization and Restriction of Chemicals (REACH) regulations, for being evaluated about their potentially dangerous properties. Accordingly, nowadays, in the UE, the commercialization of SWCNT is allowed up to 10 metric tons. Currently, the type of SWCNT registered through REACH is only that manufactured by OCSiAl [79]. Concerning the impact of using CNTS on the environment and human health, the results from some reported case studies are conflicting. In fact, while fruit fly larvae supplied with a diet containing CNTs seemed to develop normally [80], one study showed that, even if the fish otherwise appeared normal, CNTs delayed embryo development in zebrafish [81]. Lung inflammation was observed in mice when exposed to NTs, and though the inflammation collapsed within a few months, this phenomenon recalls the effect of asbestos on human

lungs [82]. The proliferation of certain human tumor cells was accelerated in the presence of CNTs [61]. Regardless, in some situations, coatings applied to the CNTs, rather than CNTs themselves, are environmental dangerous. As an example, the CNTs-based solar cells need to be coated with a cadmium-telluride mixture, which is too toxic for allowing the widespread use of such solar devices [61]. The most concern about a pervasive use of CNTs consists of their slow biodegradation, which translates in the release of tubes in the environment, their transit into our food supply, and, from there, into our bodies, with consequences unknown so far. It must however be considered that the CNTs application in electronics unlikely could be very risky because of the small volumes involved. Regardless, more investigations and regulations are needed in this field, and it would be suggestable to treat CNTs as a new chemical material, rather than as an isoform of the inert carbon [61].

### 2.1.2. Other Hurdles

Even if the potentialities of CNTs are tremendous, unsolved difficulties exist concerning the way of mass-production of CNTs-based devices. Indeed, the CNTs-based devices and prototypes created so far have been manufactured individually, at prohibitive costs for a typical consumer [83]. Researchers in industry and the academic ones are dedicating a great deal of effort to develop an automated manner of growing CNTs, endowed with uniform and predictable properties. The aim of many scientists consists of finding a method for obtaining self-assembling CNTs, as it happens when proteins self-assemble into complex superstructures. Unfortunately, since researchers do not know the mode by which such biological systems achieve self-assembly, they have no strategy to use in their pursuit of CNTs self-assembly. Until they understand a method of mass-production of CNTs, most of their possible applications will remain in the research laboratory, while silicon will continue to govern most technologies, including the computing one [83]. Even if the potentialities of CNTs are tremendous, unsolved difficulties exist concerning the way of mass-production of CNTs-based devices. Indeed, the CNTs-based devices and prototypes created so far have been manufactured individually, at prohibitive costs for a typical consumer [83]. Researchers in industry and academia are dedicating significant effort to developing an automated manner of growing CNTs, endowed with uniform and predictable properties. The aim of many scientists consists of finding a method for obtaining self-assembling CNTs, as it happens when proteins self-assemble into complex superstructures. Unfortunately, since researchers do not know the mode by which such biological systems achieve self-assembly, they have no strategy to use in their pursuit for CNTs self-assembly. Until they understand a method of mass-production of CNTs, most of their possible applications will remain in the research laboratory, while silicon will continue to govern most technologies including the computing one [83].

### 3. Approaching Bacterial Nanotubes (BNTs)

*3.1. Main Tubular Membranous Nanostructures in Bacteria*

Bacteria, depending on the species, could form different tubular membranous structures which, according to most experts, represent one of the bacterial strategies to exploit their environment. As summarized in Table 3, these structures, which can be constituted by cytoplasmic (CM) or outer membranes (OM), are labeled as nanotubes (NTs) in *B. subtilis*, *E. coli*, *Bacillus megaterium* and *Deinococcus radiodurans* [51,56], nanowires (NWs) in *Shewanella oneidensis* and bacteria of *Geobacter* genus [84,85], nanopods (NPs) in *Delftia* sp. Cs 1-4 and in *Hyperthermophilic archaea* of the genus *Thermococcus* [86–88], or outer membrane tubes (OMTs) in *Francisella Novicida* and *Myxococcus xanthus* [55,57].

**Table 3.** Tubular membranous nanostructures found in bacteria of different species.

| Nanostructure | Bacteria | Constituents | Function *<br>Occurrence (%) ** | Ref. |
|---|---|---|---|---|
| NTs | *B. subtilis*<br>*E. coli*<br>*B. megaterium*<br>*D. radiodurans* | CM | Trade of cytoplasmic<br>materials, nutrients, toxins *<br>70% ** | [51,56,59] |
| NWs | *Shewanella* genus<br>*Geobacter* genus | CM | Electrically conductive<br>appendages * | [84,85] |
| NPs | *Delfitia* sp. Cs 1-4<br>*H. archaea*<br>*Thermococcus* genus | CM | Long-distance interactions with<br>environment *<br>Deployment of OM vesicles and tubes (OMV/Ts) * | [84–86] |
| OMTs | *M. xanthus*<br>*F. novicida* | OM | Functions in pathogenesis * | [55,57] |

* Function; ** occurrence (%).

Although recent studies have reported very different finding [60], the NTs of *B. subtilis* are perhaps the best characterized example of nanostructures produced by bacteria, and several NTs for single cell have been observed in about 70% of exponentially growing cells [59].

### 3.2. Classes of BNTs Observed

Mainly, two classes of NTs have been recognized in *B. subtilis* and *E. coli*, those attached to a single cell, namely "extending nanotubes" (ENTs), and those connecting two neighbor cells, namely "intercellular nanotubes" (INTs) [51,54,89]. Although recently, different discoveries have been reported [60], several studies indicated as the specific functions of ENTs and INTs those reported in Table 4.

**Table 4.** Classes and functions of NTs in *B. subtilis*.

| Bacterial Specie | Class of NT | Location | Function | Ref. |
|---|---|---|---|---|
| *B. subtilis*<br>*E. coli* | ENTs | Attached to a single cell | ⇧ Surface area of cells<br>Contribute to nutrient uptake | [51,54,56,88,89] |
| | INTs | Connecting two cells | Conduits for transport of metabolites<br>(amino acids),<br>proteins (including toxins),<br>and non-conjugative plasmids | |

⇧ denotes high, higher, increase, major.

Further insights into BNTs are needed to have more reliable models of their functioning (which is still poorly defined). Additionally, conflicting findings are present in the literature, concerning several questions about BNTs [51,54,59,60]. However, it is widely observed that INTs can be formed both between two cells of the same bacterial species, between cells of two different bacterial species, and even between a bacterium and a eukaryotic cell. In the latter case, it has been reported that NTs are exploited by bacteria to extract nutrients from eukaryotic neighbor as observed for enteropathogenic *E. coli* [90,91]. Additionally, a better knowledge of BNTs' apparatus could be helpful to understand and fight the bacterial pathogenesis [92].

## 4. Synthesis of CNTs and Biogenesis of BNTs

*4.1. Synthesis of CNTs*

### 4.1.1. Arc Discharge (AD)

In 1991 during an AD process intended to produce fullerenes using a current of 100 amps, the formation of CNTs in the carbon soot of graphite electrodes was detected [93]. Subsequently, by the same method the first macroscopic production of CNTs was made in 1992 [94]. The high temperatures (> 1700 °C) used in this method for CNTs synthesis typically causes the CNTs' expansion, and CNTs are obtained with less structural defects in comparison with other methods [37].

### 4.1.2. Laser Ablation (LA)

This process, briefly described in Table 5, was developed by Dr. Richard Smalley and co-workers at Rice University, who produced different metal molecules by blasting metals with a laser. By replacing the metals with graphite, they succeeded in creating first MWCNTs [95], and later SWCNTs using a composite of graphite and a cobalt/nickel mixture as metal catalyst particles [96].

### 4.1.3. Chemical Vapor Deposition (CVD)

CVD is the most widely used method to produce CNTs [97] consisting of preparing first a substrate of metal catalyst NPs of nickel, cobalt, iron, or a combination [98,99] in the reactor by several possible ways, including reduction in oxides or oxides solid solutions and heating such a substrate at approximately 700 °C. Then, the growth of CNTs is triggered by bleeding into the reactor a "process gas", such as ammonia, nitrogen, or hydrogen, and a "carbon-containing gas", such as acetylene, ethylene, ethanol, or methane. CNTs develop at the locations of the metal catalyst particles, which can move at the top of the growing CNT during its growth or remain at the CNT base [100,101]. To increase the surface area for achieving higher yields of the catalytic reaction, the metal NPs can be mixed with a catalyst support, such as MgO or $Al_2O_3$. Thermal catalytic decomposition of hydrocarbon can be a promising route for the bulk production of CNTs. The most widely used reactor for preparing CNTs by CVD method is the fluidized bed reactor. The main drawback of CVD technique when support for catalyst NPs is used is the need of removing the catalyst support via an acid treatment, which sometimes could destroy the original structure of the CNTs [97,102]. Interestingly, an advanced CVD technique, namely plasma-enhanced chemical vapor deposition (PECVD) consists of generating a plasma by the application of a strong electric field during growth. In this case, the CNT growth will follow the direction of the electric field [103]. By adapting the reactor's geometry, it is possible to synthesize vertically aligned CNTs [104], whose morphology is of interest to researchers interested in electron emission from nanotubes. Among the various methods to synthetize CNTs, CVD is the most suitable for scaling-up and industrial production, due to its low cost, and the possibility of growing nanotubes directly on a desired substrate by careful deposition of the catalyst [105].

High-Pressure Carbon Monoxide (HiPco) Process

In the HiPco process developed at Rice University, SWCNTs are created from the gas-phase reaction of iron pentacarbonyl with high-pressure carbon monoxide (CO) gas. Particularly, iron NPs are produced that provide the nucleation surface for the transformation of CO into carbon during the growth of the CNTs. Experiments were made using material in quantities of milligrams to grams. Waste material was regularly removed by a professional company, and incinerated, avoiding environmental release [106].

**Table 5.** The three most applied methods to synthetize CNTs.

| Method | Process Type | Products Purity | Conditions Yields (%) | Advantages | Disadvantages |
|---|---|---|---|---|---|
| Arc discharge (AC) | The carbon in the negative electrode sublimates due to the high discharge temperatures (T) | SWCNTs short tubes Ø = 0.6–1.4 nm MWCNTs short tubes Inner Ø = 1–3 nm Outer Ø = 10 nm Medium purity | Argon/$N_2$ 500 torr T ≤ 4000 °C 20–100% | Few structural defects | Length ≤ 50 µm |
| Laser ablation (LA) | Graphite samples are vaporized in a reactor at ⇧ T by a pulsed laser in presence of an inert gas CNTs grow up on the cooler reactor's surfaces as the vaporized carbon condenses | SWCNTs 5–20 µm Ø = 1–2 nm MWCNTs Low purity | 500–1000 °C at ⇧ energy laser beam 25–1000 °C ≤ 70% | Controllable Ø by the reaction temperature | ⇧ Expensive than AC, CVD |
| Chemical Vapor Deposition(CVD) | Layered metal catalyst particles are heated in a reactor where a process gas and a carbon-containing gas are bled into | SWCNTs long tubes Ø = 0.6–4 nm MWCNTs long tubes Ø = 10–240 nm Medium-High purity | Low pressure inner gas (argon) 500–1200 °C 60–90% | ⇧ Economic and simple than AD and LA Synthesis at ⇩ temperature and ambient pressure ⇧ Yield and purity than AD and LA. Versatile in the control of CNTs structure/architecture Suitable for scale up | ⇩ Crystallinity than that by AD and LA Removal of the catalyst support |

⇧ Denotes high, higher, increase, major; ⇩ denotes low, lower, decrease, minor.

Super-Growth CVD

Super-growth CVD (SGCVD), also known as water-assisted chemical vapor deposition, developed by Kenji Hata, Sumio Iijima, and co-workers at AIST (Japan) [107], uses water into the CVD reactor, to improve the activity and lifetime of the catalyst. This process allows to obtain either dense millimeter-tall vertically aligned nanotube arrays (VANTAs) or "forests", which are aligned normally to the substrate. The synthesis efficiency is about 100 times higher than that of the LA method, and SWNT forests of 2.5 mm height can be made in 10 min. SWNT forests can be easily separated from the catalyst, yielding clean SWNT material (purity > 99.98%) without further purification, differently from the CNTs obtainable by the above-reported HiPco process, which could contain 5–35% of metal impurities; and need purification work-up through dispersion and centrifugation that damages the nanotubes [106]. The mass density of SGCVD-based CNTs is much lower than that of conventional CNT powders, probably because the latter contain metals and amorphous carbon. The vertically aligned nanotubes forests originate from a "zipping effect" caused by the surface tension of the solvent and the Van der Waals forces between the CNTs when they are immersed in a solvent and dried. VANTAs can be formed in various shapes, such as sheets and bars, by applying weak compression during the process. The packed CNTs are more than 1 mm long [108].

### 4.1.4. Other Methods
Plasma Torch (PT)

Plasma torch for producing SWCNTs was developed by Olivier Smiljanic in 2000 at Institut National de la Recherche Scientifique (INRS) in Varennes, Canada. A mixture of argon, ethylene and ferrocene is introduced into a microwave PT, where it is atomized by the plasma at atmospheric pressure, which has the form of an intense 'flame'. The fumes created by the flame contain SWNTs, metallic and carbon nanoparticles, and amorphous carbon [109,110]. The decomposing of the gas can be 10 times less energy-consuming than graphite vaporization, as made in LA or AD. SWCNTs were obtained by a modified PT method namely ITP method, implemented in 2005 by groups from the University of Sherbrooke and the National Research Council of Canada [111]. As in AD, an ionized gas is used to reach the high temperature necessary to vaporize carbon-containing substances and the metal catalysts necessary for the ensuing nanotube growth. The thermal plasma is induced by high-frequency oscillating currents in a coil and is maintained in flowing inert gas. SWCNTs with different diameter distributions can be synthesized.

Liquid Electrolysis Method (LEM)

MWCNTs can be obtained by electrolysis of molten carbonates [112], with a mechanism like that of CVD. In this case, metal ions were reduced to a metal form and attached on the cathode forming the nucleation point for the growing of CNTs. The net reaction is:

$$CO_2 \longrightarrow CNTs + O_2$$

Practically, the reactant is only greenhouse gas of carbon dioxide, while the product is high valued CNTs. This discovery represents a possible technology for carbon dioxide capture and conversions [113–116].

Natural, Incidental, and Controlled Flame Environments

Note that CNTs are commonly formed in commonplace flames produced by burning methane, [117] ethylene, [118], and benzene [119], and they have been found in soot from both indoor and outdoor air [120]. Unfortunately, these naturally occurring CNTs are highly irregular in size and quality because produced is uncontrolled conditions, and lack in the high degree of uniformity necessary to satisfy the many needs of both research and industry. Recent efforts have focused on producing more uniform carbon nanotubes in controlled flame environments [121–125]. Such methods have promise for large-scale,

low-cost nanotube synthesis based on theoretical models though they must compete with rapidly developing large scale CVD production.

### 4.2. Biogenesis of Bacterial Nanotubes: Among Sheared and Conflicting Opinions

By using B. subtilis, Dubey et al. reported that BNTs exist both as intercellular tubes connecting neighboring bacterial cells and as elongated extending tubes, connecting distant bacterial cells [53] (Table 4). In this regard, it is reported that the latter are produced by bacteria when grown at low density [53]. Additionally, the Dubey group identified a phosphodiesterase calcineurin-like protein, namely YmdB, present both in the cytoplasm and in nanotubes. YmdB is highly conserved among Gram-positive and Gram-negative bacteria and it was demonstrated that it is an essential factor for nanotubes formation and intercellular molecular exchange. Enzymatic analysis and crystal structure of YmdB evidenced that it has a metallophosphodiesterase conserved domain, that serves to hydrolyze cyclic AMP (cAMP) [126,127], which work as secondary messengers to control the social activities and the proper colony development in microbes [88,89,128,129]. Diethmaier et al. in 2014 described an additional effect of YmdB on mRNA levels of many genes, including those involved in motility, biofilm formation, and sugar utilization [127]. YmdB induces the expression of biofilm matrix genes and repress the expression of motility genes, hence controlling the switch from a motile to a multicellular sessile lifestyle [126]. Further investigations suggested that the effect of YmdB on nanotubes is independent of its role in biofilm development. Alternatively, the formation of nanotube networks might be a preceding stage in the establishment of a biofilm, providing the foundation for unhampered intercellular molecular flow between its inhabitants. Concerning the subcellular localization of YmdB, it was demonstrated that it preferentially locates to the cell circumference, often concentrated in foci-like assemblies in a frequency of approximately one per cell. Therefore, YmdB is mainly in the cytoplasmic fraction associated with the membrane. An abundance of YmdB was detected also in the nanotube fraction. Indeed, it was observed that YmdB molecules clearly coincided with nanotubes and specifically co-localize with nanotubes [54]. Collectively, it was uncovered that YmdB is associated both with the cell periphery and nanotubes. Practically, YmdB positioned to the periphery of bacterial cell hydrolyzes cAMP deriving from external sources, thus transmitting a message to produce BNTs. Furthermore, YmdB situated in the emerging nanotubes serve to feel neighbor cells and direct the tube growth toward the detected stimulus. Additionally, this external signal also regulate YmdB subcellular localization, creating high local concentrations of protein molecules, that subsequently recruit additional nanotube machinery components (Figure 2)

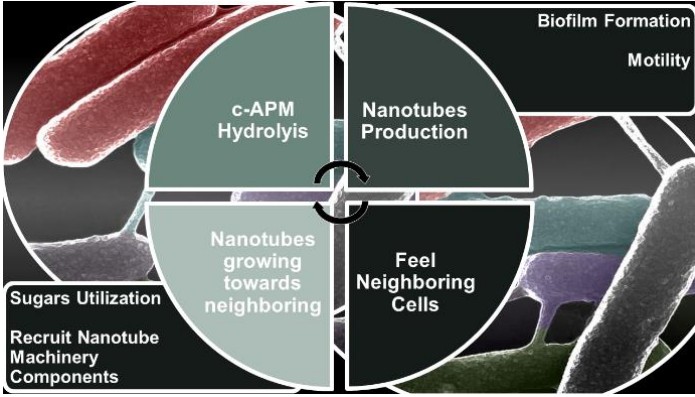

**Figure 2.** The many functions of YmdB.

Estimates of the number of bacteria cells producing NTs evidenced that 95% of the wild-type (WT) cells showed nanotubes, whereas less than 5% of mutant cells not possessing YmdB held NTs [60].

In addition to YmdB, flagellar body proteins called CORE, which are required for the flagellar export apparatus [54,59] have been reported to be necessary for BNTs formation in *B. subtilis*, and, therefore, they operate both in flagella and in BNTs assembly [54,59,130].

Particularly, Bhattacharya et al., using *B. subtilis* revealed that conserved components of CORE, dually serve for flagellum and nanotube assembly [59]. Particularly, flagellar CORE apparatus consists of a group of transmembrane proteins (Figure 3), particularly FliP, FliQ, FliR, FlhB, and FlhA in ratio 5:4:1:1:9, and of the chaperone FliO, not showed in Figure 3, since only transiently it associates with the CORE complex (Figure 3).

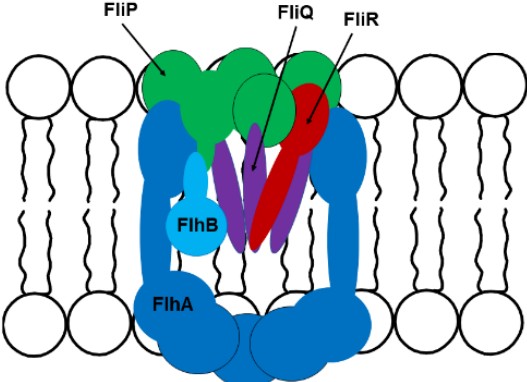

**Figure 3.** Schematic illustration of the flagellar CORE apparatus which consists of FliP, FliQ, FliR, FlhB, and FlhA (5:4:1:1:9) transmembrane proteins. The chaperone FliO has not been shown.

In this regard, the authors demonstrated that mutant bacteria, also of different species lacking CORE genes, even if gifted with other flagellar components, do not produce nanotubes and are not able to carry out intercellular molecular trafficking, thus establishing that formation of BNTs mediated by CORE is ubiquitous and is universal and phylogenetically widespread (Figure 4).

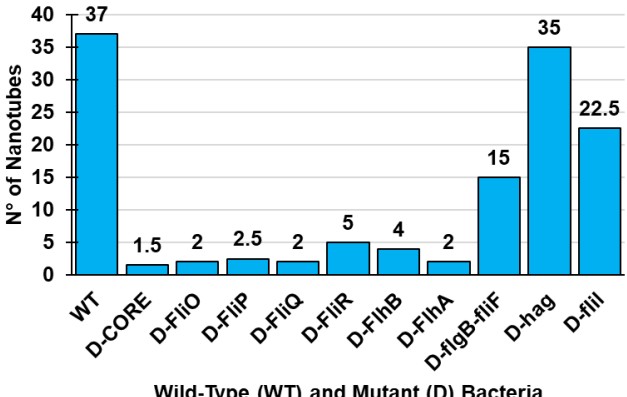

**Figure 4.** Average number of BNTs displayed per 50 cells by the indicated mutant strains following XHR-SEM analysis.

Additionally, they evidenced that the CORE components are located where the nanotube emerges analogously to what was already observed for the flagella, and that exogenous COREs deriving by different species could restore nanotube generation and functionality in bacteria lacking endogenous CORE (Figure 5) [59].

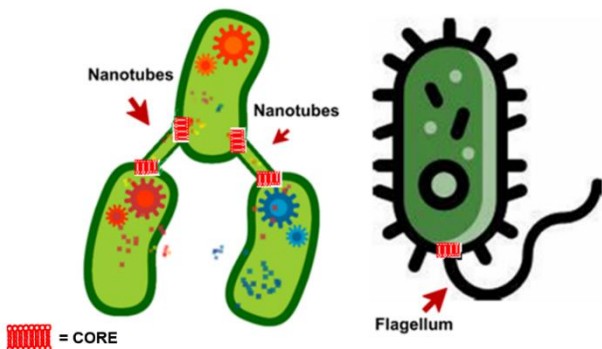

**Figure 5.** Conserved flagellar CORE components dually serve for flagella and nanotube assembly.

Recently, other genes required for NT formation were detailed using a systematic, unbiased approach [60], which allowed to identify the regulon that contains genes necessary for NTs formation. In this regard, although with no statistical significance, due to the low number of NTs produced in the tested conditions, it was determined that among the 19 sigma factors of WT *B. subtilis* the sigma factor, namely SigD, is required for NTs formation and was identified as the above-mentioned regulon, from which CORE genes depend [60]. As confirmation, exponential phase ΔsigD cells (not possessing SigD), differently from WT cells did not produced NTs within a 15 min time course experiment [60]. It was evidenced that other gene from the SigD regulon besides the CORE ones are involved in NTs formation [60]. It was demonstrated that autolysins-peptidoglycan hydrolases, such as LytE, LytF, LytB, and LytC, that open connections in the peptidoglycan net, thus allowing the insertion of afresh synthesized material for permitting surface expansion and cell separation [131], could be involved in the formation of BNTs, by weakening peptidoglycan and allowing BNTs extrusion [132]. In this regard, it was reported that in ΔsigD or ΔlytEF mutant cells the blockage of the cell wall degradation by the lack of LytE and LytF caused the delay in the appearance of NTs, while ΔlytBC mutant displays reduced NT production [132].

Recent Reports on Conditions and Requirements Promoting NT Formation

Using a combination of structured illumination microscopy (SIM) and scanning electron microscopy (SEM), it was possible to observe that, in *B. subtilis* cells (BSB1) grown to exponential phase, two types of filamentous structures were present. Particularly, several thinner filaments (diameter < 30 nm) and few thicker ones (diameter ~70 nm), identified as NTs, were observed [60]. Among the filaments believed to be NTs, elongated, flattish terminal structures were observed, but, collectively, the frequency of NTs was rather low. Only one NT approximately per 500 cells was observed, and, while a cell can have many flagella, only one nanotube, when present, is typically possessed by *B. subtilis*. Additionally, it was demonstrated that the amounts of NTs observed are sensitive both to growth conditions (solid or liquid media) and to the conditions used for preparation of the microscopic samples. Particularly, cells grown in liquid media displayed few NTs, while cells grown on solid media formed either no NTs or many NT-like structures depending on the sample preparation protocols [60]. Additionally, it was observed that the cells possessing NT structures were not in health cells. Particularly, it has been demonstrated that under non-stress conditions, NTs are uncommon; while under stress, the number of NTs increases. Indeed, it was observed that while in non-stress conditions no NT was detectable in WT *B. subtilis* cells from the exponential phase, in stress conditions by using glass slides and coverslips coated with poly-L-lysine (which negatively impacts with bacteria grow) a few NTs were observed, whose number increased when the coverslip was firmly pressed down [60]. Intriguingly, differently from Dubey et al., which indicated that NTs were formed by intact living cells [54], recently it was demonstrated that NTs structures are formed when cells are dying or even after cell death, regardless the stress applied to cause cell death, including pressure or different antibiotics. This finding established that NTs are improbable to be

involved in the uptake of nutrients or in the exchange of cytoplasmic content as proposed by previous studies [60]. Experiments demonstrated that severe damage to bacterial cells resulted both in larger amounts of dying cells and NTs production, because of the compromised cell wall and (likely) excess internal pressure. Importantly, according to recent findings, NTs do not serve as channels through which nonconjugative plasmids can be transported, but they are a sign of disintegrating cells, and further studies are necessary to assess a probable physiological role of NTs. Time-lapse microscopy experiments have demonstrated that NTs formation is a rapid process, in the order of seconds, during which the NTs use the cell plasma membrane for their extrusion. It was proposed, that since the cells are dying and start collapsing, weak spots in the cell wall may operate as channels through which NTs are extruded to release the intracellular pressure, and cardiolipin which is an integral part of the *B. subtilis* membrane might play a role in NT formation [133,134]. Indeed, it was observed that cardiolipin combined with the PmtA protein from the phytopathogenic bacterium *Agrobacterium tumefaciens* was previously reported to promote formation of tubular structures in vitro [135].

Not Only in *B. subtilis*

As observable in Table 6, in addition to *B. subtilis*, membranous tubular structures inducted by some type of stress and emerging from dying cells have been detected also in other bacterial species, such as *B. megaterium, D. radiodurans,* and *E. coli,* and in eukaryotic cells, such as macrophages. Particularly, *B. megaterium i*s rapidly inhibited by *B. subtilis* by delivering the tRNase toxin WapA and by nutrients extraction utilizing the same nanotube apparatus in a bidirectional manner, to maximally exploit potential niche resources [90].

**Table 6.** Cells reported to produce membranous tubular structures inducted by some type of stress.

| Cells | Species | Tubular Structure Type | Constituents | Inducing Stress | Physiological Role |
|---|---|---|---|---|---|
| Prokaryotic | *B. subtilis* | NTs | CM proteins CM lipids | Poly-L-lysine Pressure Different antibiotics | Hallmark of dying cells or cell death and cell disintegration |
| | *B. megaterium* | NTs | CM proteins CM lipids | N.R. | To be better investigated |
| | *D. radiodurans* | NTs | CM proteins CM lipids | Mitomycin C | Cytoplasmic constituents exchange To be better investigated |
| | *E. coli* | NTs | CM proteins CM lipids | Amino-acid deprivation | Cytoplasmic constituents exchange To be better investigated |
| | *M. xanthus* | OMTs | OM protein OM lipids | Lack of oxygen Metabolic inhibitors | Hallmark of ghost cells |
| | *F. novidica* | OMTs | OM protein OM lipids | Amino-acid deprivation Macrophage infection | Hallmark of ghost cells |
| | *F. tularensis* | OMTs | OM protein M lipids | Amino-acid deprivation Macrophage infection | Hallmark of ghost cells |
| Eukaryotic | Macrophages | TNTs | F-actin Proteins | HIV-1 *Mycobacterium tuberculosis* | Channels for cell-to-cell communication |

Interactions within and between *Acinetobacter baylyi* and *E. coli*, in which two distant bacterial species can connect each other via membrane-derived nanotubes and use these to exchange cytoplasmic constituents, such as amino acids, have been reported when auxotrophy-causing mutations were induced [56]. Tubular structures, termed outer membrane tubes (OMTs), induced by stress, such as lack of oxygen or addition of metabolic inhibitors, have been observed in *M. xanthus* [136,137]. OTMs contain outer membrane (OM) proteins and lipids and no other cytoplasmic material and are not involved in the intercellular transfer of OM proteins, which instead depends on the TraAB system and direct cell-to-cell contact. On the contrary, Wei et al. [57] reported that OMTs are predominantly associated with ghost cells. Additionally, OMTs were also found in *F. novidica* and *F. tularensis* when subjected to conditions of stress induced by amino-acids deprivation or during infection by macrophages [57,138]. Nevertheless, these structures, being formed from the OM, do not have to cross the cell wall and, consequently, the mechanism of their formation is different from that of production of BNTs in *B. subtilis*. Finally, NTs-like structures were observed in *D. radiodurans* when stressed with mitomycin C [139]. Various types of tubular structures have also been reported for eukaryotic cells and mitochondria, in some cases, paradoxically induced by the presence of bacteria. Tunneling nanotubes (TNTs) of macrophages are an example [140,141]. TNTs are long-range membranous F-actin containing tube-like structures, that have been classified into two types based on their thickness and the presence or absence of microtubules [142], whose formation can be inducted by HIV-1 virus whose formation is stimulated by coinfection with *M. tuberculosis* [143]. TNTs, however, appear to be distinct from bacterial NTs by the presence of a protein scaffold and appear to be channels for cell-to-cell communication. Collectively, it can be assumed that *B. subtilis* NTs are a trait of dying cells, or cell death, and are involved in the final cell collapse. In other bacterial species, similar structures should be studied with utmost care before attributing physiological roles to them.

## 5. Architecture and Other Features of CNTs and of BNTs

### 5.1. Structure of CNTs

CNTs merit the "nano" prefix since they can be as small as 10 atoms across, i.e., 1/50,000th the diameter of a human hair. As previously reported, carbon is the ingredient from which CNTs are generated by the methods described in Section Multi-walled Nanotubes 3.1.

Pure carbon can form many structures and CNTs represent only one of the many structures that pure carbon can produce. The tetrahedron crystalline structure of carbon is known as diamond (Figure 6a). The planar structure in which the carbon atoms are linked to form hexagons (hexagonal lattice of carbon) is known as graphite or graphene (Figure 6b) [144]. The geodesic sphere structure in which the hexagons made of pure carbon atoms are linked to form a sphere is known as fullerene (Figure 6c). Fullerenes are also known as "buckyballs" from the name of F. Buckminister Fuller, who designed the first geodesic dome [145]. When the sheet of carbon atoms linked to form hexagons (graphene) is rolled up to form a cylinder, produces the typical structure of NTs (Figure 6d).

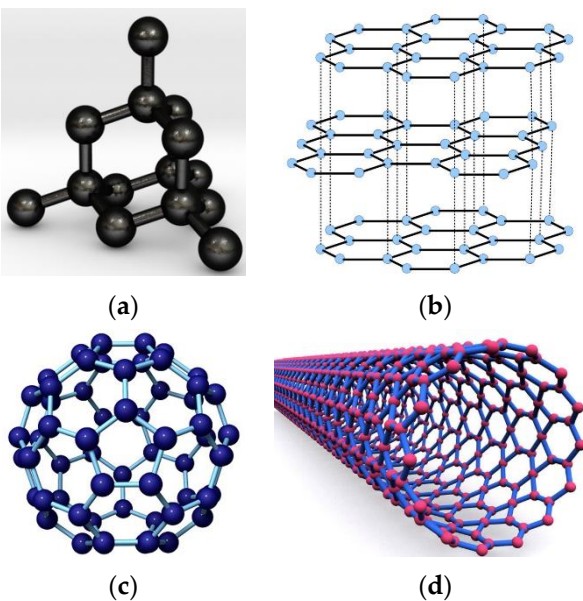

**Figure 6.** Structure of diamond (**a**), graphite (**b**), fullerene (**c**), and nanotube (**d**). These images by unknown author are licensed under CC BY.

CNTs can, in turn, assume different structures (Figure 7). Not naturally formed SWCNTs (Figure 7, left side) are single-layered tubes, while MWCNTs are CNTs formed naturally. They can consist either of a single sheet rolled several times like a roll of packaging paper (Parchment model), (Figure 7, central image) or of several concentric CNTs bound together by weak Van der Waals interactions in a tree ring-like structure (Russian Doll model) [145]. Although not identical, these latter CNTs are very similar to Oberlin, Endo, and Koyama's nanostructures, which are parallel and long straight carbon layers arranged cylindrically around a hollow tube (Figure 7, right side) [145].

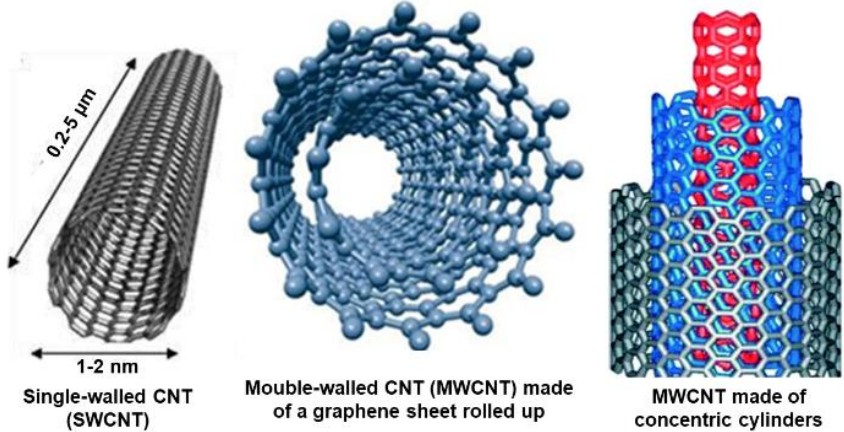

**Figure 7.** From the left: structures of a SWCNT, of a MWCNT made of a single graphene sheet rolled up and of a MWCNT made of more than two graphene cylinders one inside the other. These images by unknown author are licensed under CC BY.

The atomic structure of CNTs is very stable, but the structure of SWCNT is more stable than of the MWCNT. No intrinsic limit exists on how long a tube can become [146]. Note that are considered CNTs, tubes with an unspecified carbon-wall structure and diameters less than 100 nm [147]. Indeed, the length of a carbon nanotube is often not reported but typically, it is much larger than its diameter [146]. Table 7 collects the physical limits of CNTs produced so far.

**Table 7.** Physical limits of CNTs produced so far.

| Physical Property | Largest | Smallest | Ref. |
|---|---|---|---|
| Diameter | N.R. | 0.30 nm (MWCNT) [1]<br>0.43 nm (SWCNT) | [148] |
| Length | 0.5 m [2] | Cycloparaphenylenes (2.49 Å) [3] | [149]<br>[150] |
| Density | 1.6 g/cm * | N.R. | [151,152] |

[1] Armchair structure (see later); [2] [149]; [3] [150]; * ohmic conductivity, lowest resistance of 22 kΩ.

SWCNTs are not produced as easily as MWCNTs are, and they can exhibit behavior not found in their MW counterparts. Scientists, while continuing to explore applications and production techniques of MWCNTs, are looking for practical ways to mass-produce SWCNTs. Additionally, we note that when CNTs are formed by two or three graphene cylinders one inside the other, they are referred to as double- and triple-walled carbon nanotubes (DWCNTs and TWCNTs). Figure 8 shows the typical structure of a DWCNT.

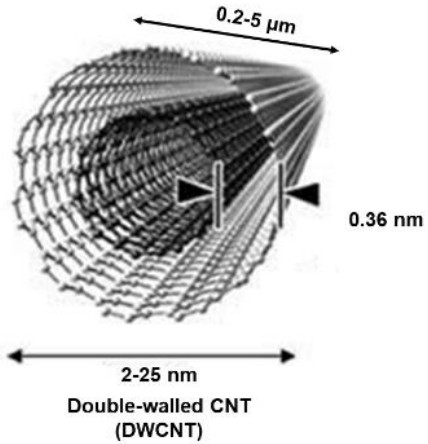

**Figure 8.** Structures of a DWCNT. This image by unknown author are licensed under CC BY.

### 5.1.1. From Traditional CNTs-Structures to CNTs-Base Derivatives or Particular Morphologies

Generally, SWCNTs have better properties than MWCNTs, even if it is more difficult to disperse SWCNTs to obtain composite materials. In this regard, CNTs can join both with other CNTs and with graphene and junctions between two or more CNTs have been widely discussed theoretically [153,154], while those between nanotubes and graphene have been considered theoretically [155] and studied experimentally [156] (Table 8). Junctions between CNTs are often observed in NTs prepared by AD, as well as by CVD methods, and the electronic properties of such junctions were first studied theoretically by Lambin et al. [157]. Particularly, they imagined that a connection between a metallic tube and a semiconducting one could, therefore, form a component of a nanotube-based electronic circuit [157]. Differently, nanotube-graphene junctions represent the basis of pillared graphene, in which parallel graphene sheets are divided by short NTs, thus representing a class of three-dimensional (3D)-carbon nanotube (3D-CNTs) constructions [158]. Recently, several studies have highlighted the feasibility of using CNTs as building blocks to fabricate three-dimensional macroscopic (>100 nm in all three dimensions) free-standing, porous, all-carbon scaffolds, possessing macro-, micro-, nano-structured pores, as well as tailorable porosity usable for the fabrication of the next generation of energy storage, supercapacitors, field emission transistors, high-performance catalysis, photovoltaics, biomedical devices, implants, and sensors [159–161].

**Table 8.** Possible Associations between the different known allotropes of carbon and particular morphologies of CNTs.

| Associations Morphologies | Type of Interaction Arrangement | Possible products | Applications/Properties | Ref. |
|---|---|---|---|---|
| CNTs/CNTs * | Junctions s | Metallic CNTs/semiconducting CNTs | Component of NTs-based electronic circuits | [153,154,157] |
| CNTs/graphene * | Junctions | Pillared graphene (3D-CNTs) | Energy storage Supercapacitors Field emission Transistors High-performance catalysis Photovoltaics Biomedical devices Implants Sensors | [158–161] |
| 2 CNTs/fullerenes * | Covalent Bond | Fullerene-like nanobuds | Field emitters | [162] |
| CNT/fullarene * | Entrapment | Carbon peapods (CPPs) | Heating devices Irradiating devices Oscillators | [163–166] |
| Doughnut shape ** | CNTs twisted into a toroid (Annulus shape). | Nanotori (NTRs) | ⇑ magnetic moment ⇑ stability | [167,168] |
| GFs/MWCNTs * | GFs integrated MWCNTs | Graphenated carbon nanotubes (GCNTs) | ⇑ surface area 3D-framework ⇑ total charge capacity per unit of nominal area | [169,170] |
| quasi-1D carbon structures ** | Stacking microstructure of graphene layers | Cup-stacked carbon nanotubes (CSCNTs) | Semiconductors | [171] |

* Associations; ** particular morphologies; ⇑ denotes high, higher, increase, major; ⇓ denotes low, lower, decrease, minor.

Carbon Nanobud Structures (CNBs)

CNBs are nanostructures that combine two CNTs and fullerenes, forming fullerene-like "buds" which bind covalently the underlying CNTs, thus providing hybrid materials possessing properties of both fullerenes and CNTs. Particularly, they have improved mechanical properties and have been found to be exceptionally good field emitters [162].

Carbon Peapods (CPPs)

CPPs are other carbon-based nanocomposites in which fullerene moieties are blocked inside a CNT. CPPs can possess appealing heating and irradiation capacities and magnetic properties, and can be also applied as an oscillator during theoretical investigations and predictions [163–166].

Nanotori (NTRs)

A nanotorus (NTR) is a carbon nanotube twisted into a toroid (annulus shape). NTRs could have many unique properties, such as very large magnetic moments and thermal stability, mainly based on the radius of the toroid and of the tube [167,168].

Graphenated Carbon Nanotubes (GCNTs)

GCNTs are hybrid materials that bring together graphitic foliates (GFs) grown along the sidewalls of MWCNTs or bamboo style CNTs, in which the density of the foliate

depends on the deposition conditions (temperature and time). The obtainable structures can have either a few layers of graphene (< 10) or thicker ones, with more graphite-like characteristics [169]. The integrated graphene-CNT structures have a high surface area three-dimensional framework. Additionally, by depositing a high density of graphene foliates along the length of aligned CNTs, the total charge capacity per unit of nominal area can be significantly improved, as compared to other carbon nanostructures [170].

Cup-Stacked Carbon Nanotubes (CSCNTs)

CSCNTs differ from other quasi-1D carbon structures, which normally behave as quasi-metallic conductors of electrons. CSCNTs exhibit semiconducting behavior because of the stacking microstructure of graphene layers [171].

### 5.1.2. More Insights in the CNTs' Structure: Relationships Structure/Properties

We suggest that the process by which the atoms bond to form CNTs is not completely understood, and the study of nanotubes continues to be a cutting-edge field, where new discoveries are being made regularly. However, while, as above reported, CNTs can exist as SWCNTs, DWCNTs, TWCNTs, and MWCNTs, it is also accepted that the structure of an ideal (infinitely long) SWCNT is that of a regular hexagonal lattice drawn on an infinite cylindrical surface, whose vertices are the positions of the carbon atoms. In this regard, among the possible structure that a SWCNT can assume the armchair, zigzag, and chiral configurations (or structural designs) are extensively recognized (Figure 9).

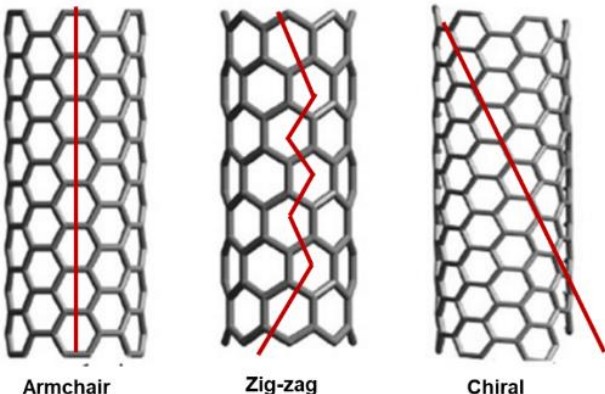

**Figure 9.** Armchair, zigzag, and chiral structures of SWCNTs. This images by unknown author are licensed under CC BY.

The CNTs model depends on the way the graphene is packaged into a cylinder. Importantly, the structural design of the CNTs has a direct effect on the nanotube's properties, and mainly on the mechanical and electrical ones. In some case, MWCNTs could have similar properties to SWCNTs. Regardless, in MWCNTs, the outer walls can protect the inner carbon tubes from chemical interactions with outside materials.

Structure-Related Mechanical Properties

Carbon nanotubes are the greatest nanomaterials discovered so far, in terms of elastic modulus and tensile strength. Generally, CNTs have a higher tensile strength than steel and Kevlar, coming from the $sp^2$ bonds between the individual carbon atoms. Particularly, a MWCNT was found to have a tensile strength of 63 gigapascals (9,100,000 psi) [172], while SWCNTs revealed to have strengths of up to ≈100 gigapascals (15,000,000 psi) [173]. On the other hand, CNTs since elastics, due to their hollow structure and high aspect ratio, tend to undergo buckling when placed under compressive, torsional, or bending stress [174]. The first transmission electron microscope observation of radial elasticity suggested that even Van der Waals forces can deform two adjacent nanotubes. Later, the radial elasticity of MWCNTs and SWCNTs was quantitatively measured by several groups,

thus demonstrating that CNTs are, in fact, very soft in the radial direction. You can press on the tip of a CNT and curve it without damaging irreversibly its structure. Indeed, the nanotube will return to its original shape when the force will be removed [61]. The elasticity of a CNT has a limit, and very strong forces could permanently deform its shape. Defects in the structure of the CNTs, occurring from atomic vacancies or rearrangements of the carbon bonds, can cause the emergence of weak points in a small segment of the CNT, which, in turn, causes the tensile strength of the entire nanotube to weaken. The total tensile strength of a CNT depends on the strength of the weakest segment in the tube [61].

Structure-Related Electrical and Thermal Properties

The structure of CNTs affect the conductive potency of the nanotube, both in terms of electrical and thermal conductivity. When the structure of atoms in a CNT minimizes the collisions between conduction electrons and atoms, a CNT shows highly conductive power. Particularly, CNTs could be either metallic or semiconducting along the tubular axis. All armchair structured CNTs are metallic, while nanotubes differently structured are semiconducting [175]. It must be considered that the rolling of the graphene sheet which forms the CNT can be done at different angles and curvatures resulting in different nanotube semiconducting properties. The diameter typically changes in the range 0.4–40 nm (i.e., "only" ~100 times), but the length can vary from 0.14 nm to 55.5 cm (i.e., (~100,000,000,000 times) [176]. The CNTs aspect ratio, or the length-to-diameter ratio, can reach the value of 132,000,000:1 [177]. Consequently, all the properties of the CNTs relative to those of typical semiconductors are extremely anisotropic (directionally dependent) and tunable. It must be considered that curvature effects in small-diameter tubes can strongly influence their electrical properties. Armchair-structured metallic nanotubes can carry an electric current density of $4 \times 109$ A/cm$^2$, which is more than 1000 times greater than those of metals, such as copper [146]. Doping in CNTs semiconductors differs from that of bulk crystalline semiconductors from the same group of the periodic table (e.g., silicon). Substitution of carbon atoms in the nanotube wall by boron or nitrogen dopants leads to p-type (excess of positive (p) electron charge) and n-type (excess of negative (n) electron charge) semiconductors, respectively, as would be expected in silicon. However, some non-substitutional (intercalated or adsorbed) dopants introduced into a CNT, such as alkali metals and electron-rich metallocenes, result in n-type conduction. By contrast, $FeCl_3$ or electron-deficient metallocenes function as p-type dopants. The strong bonds between carbon atoms characterizing CNTs, allow them to withstand both higher electric currents and higher temperatures than copper. Because of this, CNTs have been shown to be also very good thermal conductors. Particularly, single SWCNTs are expected to be very good thermal conductors along the tube, while good insulators lateral to the tube axis [178]. Networks composed of nanotubes demonstrate different values of thermal conductivity, from the level of thermal insulation with the thermal conductivity of 0.1 W·m$^{-1}$·K$^{-1}$ to a conductivity of 1500 W·m$^{-1}$·K$^{-1}$ [178,179], due to the thermal resistance which the system could have caused by the presence of misalignments, impurities, and other defects. Crystallographic defects strongly affect the tube's thermal properties. Phonon transport simulations indicate that substitutional defects, such as nitrogen or boron, will primarily lead to scattering of high-frequency optical phonons. However, larger-scale defects such as Stone–Wales defects cause phonon scattering over a wide range of frequencies, leading to a greater reduction in thermal conductivity [180].

*5.2. Architecture, Dynamics, Molecular Components, and Other Concerning BNTs*

5.2.1. Ten Years of Studies and Discoveries

According to Dubey et al. which studied the dynamics, architecture, and molecular components of BNTs utilizing *B. subtilis* as a model organism, as already mentioned in Table 4, at low cell density, nanotubes exhibit remarkable complexity, existing as both INTs and ENTs. Both are composed of constricted chains of sequential beads of the lipidic cytoplasmic membrane sheltering a continuous lumen that contain mainly the

phosphodiesterase protein YmdB essential for nanotube formation and functionalities [54] (Figure 10).

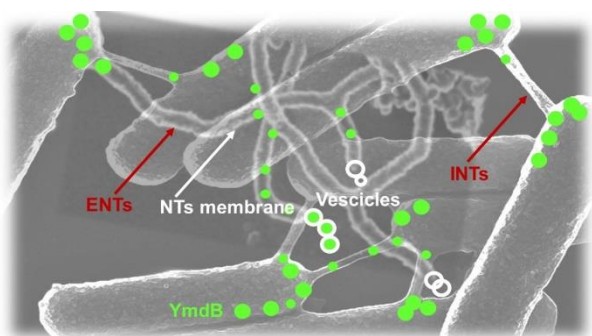

**Figure 10.** Example of a network of intercellular and elongated extending tubes.

Particularly, the authors, after having established the existence of INTs connecting two neighboring bacterial cells grown at high density [51], observed that single cells grown at low density and with no neighbor cells produce anyway elongated tubular protrusions frequently arranged in a "root-like" fashion. Since such structures determine a dramatic extension of the surface of bacterial membrane, were labeled extending nanotubes (ENTs). ENTs share the same morphology of INTs and when encounter other cells, could interconnect the two partners, displaying manifestation comparable with the nanotubes produced at high cell density [54]. In their work of 2011, Dubey and co-workers provided evidence that BNTs are sensitive to detergent treatment, have a composition like that of bacterial membranes, present the characteristic lipid bilayer, and their tubular nature depends just on the lipid composition of the membrane [51]. More recently, they demonstrated that both INTs and ENTs lack cell wall layers [54]. Typically, BNTs appear as a chain of consecutive constricted segments having an uninterrupted lumen. Sometimes these constrictions were less evident, and the tubes appeared more homogeneous in width, while in other cases the constriction sites were prominent and free vesicles seemed to be released from the nanotubes [54]. Estimations based on different analytical techniques assessed that the tube width can range from 50 to 70 nm but at constricted sites nanotube width was reduced to 20 nm, while the length of individual nanotube segments, as defined by two constriction sites, was found around 100 nm. However, following NTs formation, Dubey observed that a single nanotube could extend to approximately 15 μm within 15 min and after 70 min, could reach an approximate length of 57 μm with a corresponding surface area of 9 μm$^2$, which is almost three times larger than the calculated surface of a typical *B. subtilis* cell. According to Dubey, the tremendous increase in cell surface area could support the assumption that ENTs serve to scavenge for nutrients, while the observed rapid nanotube movements, on a timescale of milliseconds, supports the premise that nanotubes could serve to explore the surroundings. Additionally, using the small fluorophore calcein acetoxy-methyl-ester (AME) (623 Da), it was demonstrated its presence within tubes and was found its rapid exchange among neighbor bacteria cells [51], thus further supporting the assumption that nanotubes serve for the sharing of cytoplasmic metabolites, nutrients, toxins, cytoplasmic proteins, and even plasmids across the connecting structure.

### 5.2.2. The Conflicting Recent Reports

Intriguingly, as noticed in other parts of the present work, also concerning the architecture, dynamics, and molecular components of BNTs, in a recent study by Pospíšil et al., discoveries in strong contrast with those of Dubey et al. [51,54] and of other researchers [56–59,67,68] have been reported [60]. Accordingly, BNTs are produced exclusively from dying cells, likely because of biophysical forces, and their emergence is extremely rapid, happening within seconds by disassembling the cell membrane. BNTs are a structure emerging post-mortem, which participate in the cell degeneration and are

unlikely to be involved in cytoplasmic content exchange between live cells. Indeed, it was demonstrated that the transfer of non-conjugative plasmids was not associated with NTs function but was exclusively dependent on the cell ability to take up exogenous DNA, thus ruling out any NT involvement. Accordingly, BNTs are not made of membrane lipids, but it was speculated that they might be derived from the disrupted extracellular matrix, whose disruption could be caused by imprinting of the bacteria on electron microscopy grids (EMG) and/or subsequent manipulations with the grids operated for performing the experiments. Additionally, it was even assumed that although NTs were reported to connect cells, and in some of the images recently reported such cell pairs were detected, the authors believe that these connecting NTs are artifacts of the microscopy techniques. Notably, the NTs likely emanate from one cell and their involvement with other neighboring bacterial cells creates the illusion of a connection. In any case, the occurrence of NTs is scarce and these connecting NTs are even less frequent. Furthermore, it was confuted that the transport of metabolites, such as amino acids, could be a contact-dependent phenomenon mediated by BNTs [56,68]. On the contrary, it was asserted the metabolites taken up by the auxotrophic cells in the study of Pospíšil et al. were released from nearby lysed cells [60]. Figure 11 shows a model of BNTs formation according to Pospíšil et al. [60].

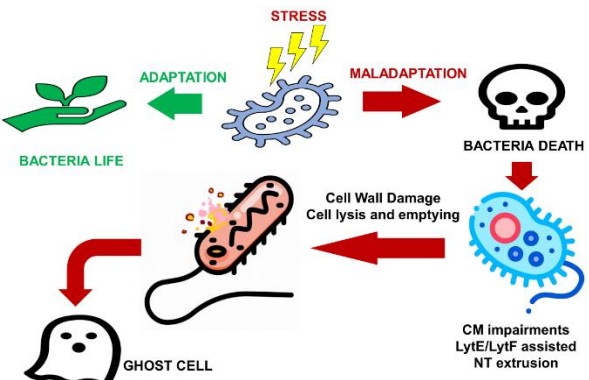

**Figure 11.** Model of BNTs formation (CM = cytoplasmic membrane; LytE and LytF = autolysins.

Particularly, cells stressed by different factors, such as pressure, or ampicillin treatment, either adapt and survive or, when the stress is too severe, die. During cell death, the CM becomes compromised and tubular structures made of the same CM are extruded from dying cells. Autolysins, mainly LytE and LytF, that localize at the cell poles weaken the cell wall and facilitate the BNT extrusion. Eventually, the cell contents are emptied and a hollow husk—the ghost cell—remains. No physiological role for these NTs has been detected, so far.

## 6. Conclusions

In this article, the main knowledge and findings gathered so far about CNTs and BNTs have been reviewed. Providing an intriguing comparison between them, it was evidenced that so different tube-like structures, with so different functions and/or applications, curiously share the same name of nanotubes and have both nanosized dimensions. Additionally, as frequently happens when nanomaterials are approached, also in the case of CNTs and BNTs, conflicting opinions exist. In fact, while researchers in the field of CNTs are excited by their potentialities, work incessantly to improve their quality and support their more extensive application, considering CNTs the material of the future, many people are sceptic concerning their use. Similarly, contrasting opinions exist also in the field of BNTs, this time concerning their actual function, finalization, composition, and even their real existence. Surely, further extensive studies to clarify many outstanding questions about both CNTs and BNTs are needed, thus establishing that the world of "nano-things" is for the most part still unexplored. However, it must be recognized that CNTs, although the still not clear toxicity, are ubiquitously exploitable for improving the quality of human

life, thus resulting of global interest. On the other hand, a greater knowledge about BNTs biosynthetically produced by prokaryotes, whose functions are not still fully clarified, could be crucial to better understand the mechanisms of pathogenesis and combat the phenomenon of resistance, which is a global concern whose solution is the daily challenge of microbiologists worldwide.

**Author Contributions:** Conceptualization, investigation, writing—original draft preparation, supervision, project administration, S.A. Writing—review and editing S.A. and G.C.S. All authors have read and agreed to the published version of the manuscript.

**Funding:** This research received no external funding.

**Data Availability Statement:** Not applicable.

**Conflicts of Interest:** The authors declare no conflict of interest.

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
