# Peer review of "Nanotubes: Carbon-Based Fibers and Bacterial Nano-Conduits Both Arousing a Global Interest and Conflicting Opinions"

_fibers, doi:10.3390/fib10090075_

Round 1

Reviewer 1 Report

Authors did good reviews on the carbon based fibers and the applications. Only some comments need to be improved, before considering the publications.

1.    The text and background of Figure 2 look chaotic and need to be revised

2.    On line 472, why did the author give a subtitle, Not Only in B. subtilis?

3.    Figure 6 needs to be modified, a clearer source, or redraw it.

4.    The topic of this review is about carbon based fibers, what is the main Global Interest and Conflicting Opinions, need to be covered more in the conclusions.

Author Response

Comments and Suggestions for Authors

Authors did good reviews on the carbon based fibers and the applications. Only some comments need to be improved, before considering the publications.

The authors are very thankful to the Reviewer of his positive comments and his appreciations.

  1. The text and background of Figure 2 look chaotic and need to be revised

The Reviewer is right. Figure 2 has been remade and now it appears much clearer.

  1. On line 472, why did the author give a subtitle, Not Only in B. subtilis?

We wanted to specify that even if nanotubes are studied mainly in B. subtilis, they can be produced also by other bacteria. Anyway, we agree with the Reviewer that “Not Only in B. subtilis” is useless and it has been removed (line 207).

  1. Figure 6 needs to be modified, a clearer source, or redraw it.

As asked, Figure 6 has been redrawn. We hope that the new Figure is good for the Reviewer.

  1. The topic of this review is about carbon based fibers, what is the main Global Interest and Conflicting Opinions, need to be covered more in the conclusions.

We make kindly note to the Reviewer that our review is not only about carbon nanotubes, but it reviewed also another intriguing type of nanotubes, that except for the name, are dramatically different from tubes made of carbon, being made mainly of membrane lipids. Anyway, we have satisfied the Reviewer request, by explaining what the main global interest and conflicting opinions are, in the Conclusions section (lines 591-604).

Reviewer 2 Report

Dear authors

Congratulations for the complete review work, the extensive bibliography, use, and the efforts summarized in a very ordered and understandable paper.

Nevertheless, it is a pity, that the only formula referring to a reaction is poorly edited (line 334). My only recommendation is to edit in the correct format this reaction, as well as some alignments in the Tables.

The size of some pictures is too big. Please, consider to readjust them

In my opinion, the authors made and built a very complete framework up. Everything related to the structural composition of nanotubes, as well as their possible applications is considered from my experience in the field.
The bibliography is very complete. I've checked some of them because I was not aware of.
Nevertheless, in my opinion, some pictures are too big to be included in the text, especially in parallel format. And the only chemical formulae is not well edited.
These have been my only concerns about the work

Author Response

Dear authors

Congratulations for the complete review work, the extensive bibliography, use, and the efforts summarized in a very ordered and understandable paper.

We thank a lot the Reviewer for his appreciations about our review.

Nevertheless, it is a pity, that the only formula referring to a reaction is poorly edited (line 334). My only recommendation is to edit in the correct format this reaction, as well as some alignments in the Tables.

We thank the Reviewer for his suggestions. The reaction has been edited. Particularly, it was obtained by using ChemDraw Ultra 7.0 software used worldwide to draw chemical structures and reactions schemes. Alignments in Tables have been checked.

The size of some pictures is too big. Please, consider to readjust them

The reviewer is right. All Figures have been resized.

In my opinion, the authors made and built a very complete framework up. Everything related to the structural composition of nanotubes, as well as their possible applications is considered from my experience in the field.

The bibliography is very complete. I've checked some of them because I was not aware of.

Nevertheless, in my opinion, some pictures are too big to be included in the text, especially in parallel format. And the only chemical formulae is not well edited.

These have been my only concerns about the work.

The concerns of the Reviewer have been met.

Reviewer 3 Report

The submission by Alfei & Schito entitled, Nanotubes: Carbon-Based Fibers and Bacterial Nano-Conduits Both Arousing a Global Interest and Conflicting Opinions" attempts to recount various aspects of NTs from a global interact and conflicting opinions perspective. As a review, this article cannot be accepted. The overall findings are imprecise, vague, subjective, and not helpful to the fiber community. It is not per se a review in the technical sense, but more an editorial attempting to synthesize aspects of NT's safety, production, and utility. If the Authors work on making it into an editorial or perspectives (shortening it substantially), it would be much more meaningful.

Author Response

We are very sorry to learn the completely negative opinion of the Reviewer about our review manuscript. To meet the Reviewer concerns, we have carefully checked all our paper and checked what reported in literature and we have not found great imprecisions. We think that a subjective contribution by the authors in a review could be an added value and not a detrimental factor, which make the work original. We think that reduce our work to an editorial or perspective would cause the loss of essential information.

Round 2

Reviewer 1 Report

This Paper could be accepted for publications now, for the careful revision.

Author Response

This Paper could be accepted for publications now, for the careful revision.

The authors are very thankful to the Reviewer for his positive comments and his appreciations, concerning the work of revision made by us to address his previous comments.

Reviewer 3 Report

I am very insulted by the Authors' comments. They do nothing to assuage the lack of meaningful data in this document. By ascribing three other reviews versus mine is a meaningless gesture toward acceptance. Unless the Authors carefully consider and address my comments, I cannot agree to this review. It is not a helpful contribution to the general community based on its current form. Again, it is more of a melange of comment, editorializing, and data. I have reviewed thousands of submissions in all forms and do not find this particular one helpful to the fiber community.

Author Response

We apologise to the Reviewer if he felt insulted by us, it was not in our intention. So, in this second-round revision, we have tried to find a compromise with him. To this end, we have re-checked carefully all manuscript and made changes to reduce what we have found could be imprecise, vague, and subjective findings. We hope the Reviewer will appreciate our efforts.